# SOD2 Gene Variants (rs4880 and rs5746136) and Their Association with Breast Cancer Risk

Martha P. Gallegos-Arreola [1],*, Ramiro Ramírez-Patiño [2], Josefina Y. Sánchez-López [1],
Guillermo M. Zúñiga-González [3], Luis E. Figuera [1], Jorge I. Delgado-Saucedo [4], Belinda C. Gómez-Meda [5],
Mónica A. Rosales-Reynoso [3], Ana M. Puebla-Pérez [4], María L. Lemus-Varela [6], Asbiel F. Garibaldi-Ríos [7],
Nayely A. Marín-Domínguez [1], Diana P. Pacheco-Verduzco [1] and Emaan A. Mohamed-Flores [8]

[1] División de Genética, Centro de Investigación Biomédica de Occidente (CIBO), Instituto Mexicano del Seguro Social (IMSS), 44340 Guadalajara, Mexico
[2] Departamento de Medicina y Ciencias de la Vida, Centro Universitario la Ciénega, Universidad de Guadalajara, 47810 Ocotlán, Mexico
[3] División de Medicina Molecular, CIBO, IMSS, 44340 Guadalajara, Mexico
[4] Laboratorio de Inmunofarmacología, Departamento de Farmacología, Centro Universitario de Ciencias Exactas e Ingenierías, Universidad de Guadalajara, 44100 Guadalajara, Mexico
[5] Departamento de Biología Molecular y Genómica, Instituto de Genética Humana "Dr. Enrique Corona Rivera", Centro Universitario de Ciencias de la Salud, Universidad de Guadalajara, 44100 Guadalajara, Mexico
[6] Departamento de Neonatología, Hospital de Pediatría, UMAE, CMNO, IMSS, 44340 Guadalajara, Mexico
[7] Doctorado en Genética Humana, IGH, CUCS, Universidad de Guadalajara, 44100 Guadalajara, Mexico
[8] Hospital de Gineco-Obstetricia, UMAE, CMNO, IMSS, 44340 Guadalajara, Mexico
* Correspondence: marthapatriciagallegos08@gmail.com; Tel.: +52-33-36170060 (ext. 31936)

**Abstract:** The superoxide dismutase (SOD) is the principal antioxidant defense system in the body that is activated by a reactive oxygen species. Some variants of the *SOD2* gene have been associated with cancer. The rs4880 variant was determined by PCR real-time and the rs5746136 variant by PCR-RFLP in healthy subjects and in breast cancer (BC) patients. The rs4880 and rs5746136 variants were associated with BC susceptibility when BC patients and the control group were compared for the *CT*, *TT*, *CTCC*, and the T alleles ($p < 0.05$). The *CT* genotype of the rs4880 variant showed significant statistical differences in patients and controls aged $\leq 45$ years old, and with hormonal consumption ($p < 0.05$). The rs4880 variant was associated with BC patients with *CTTT* genotype and obesity, the presence of DM2-SAH, and a non-chemotherapy response ($p < 0.05$). Additionally, the rs5746136 variant was associated with susceptibility to BC with Ki-67 ($\geq 20\%$), luminal A type BC, and a chemotherapy partial response ($p < 0.05$) in BC patients who carry *TT*, *TC*, and *CTTT* genotypes, respectively. The haplotype *T/T* (OR 1.98; 95% CI 1.20–3.26, $p = 0.005$) was observed to be a risk factor for BC. The rs4880 and rs5746136 variants in the SOD2 gene were associated with BC susceptibility.

**Keywords:** breast cancer; ROS; SOD2; variant; rs4880; rs5746136





## 1. Introduction

Breast cancer (BC) is the most common gynecological cancer in women, and represents a health problem in Mexico and around the world due to its high incidence and mortality rates [1,2]. The accumulation of genetic and epigenetic changes that occur in the ducts and lobules of breast tissue may influence cancer development [3]. There are well-documented studies that have shown the relationship between the development of cancer and reactive oxygen species (ROS). Among the main known characteristics are the fact that they are highly reactive and are produced by ultraviolet rays, pharmaceutical drugs and different diseases, and ROS activate inflammatory cells [4–6]. Regarding cancer, ROS are considered tumor promoters because of the increasing oxidative DNA damage; they can promote the transformation of oncogenes and increase cell proliferation, survival, and migration [4–7]. All neoplastic changes occur with an increase in the rate of cell proliferation and the

suppression of apoptosis [8]. Estrogens and their metabolites generate ROS. Additionally, the xenobiotic accumulation can damage DNA, leading to the initiation or promotion of cancer [3].

When the production of ROS is excessive in the body and the antioxidant defense capacity is inefficient, oxidative stress is generated. This can cause severe health problems [4–7]. The superoxide dismutase (SOD) is the principal antioxidant defense system in the body which is activated by ROS; it is distributed in the mitochondria, peroxisomes, and cytoplasm [4].

Different isoforms of SOD have been identified in mammals: SOD1, SOD2 (MnSOD; EC 1.15.1.1), and SOD3. SOD2 uses manganese as a cofactor. It is located in the mitochondria of aerobic cells and neutralizes ROS, which are produced in the mitochondria during the respiratory chain. This transforms them into $H_2O_2$, which is metabolized by a catalase into molecular oxygen and water. Gene *SOD2* is located in the chromosomal region 6q25, is structured by five exons and four introns, presents *GC*-rich regions, and the promoter lacks the *TATA* or *CAAT* box. The $3'$ region has a binding site for the transcription regulation factor NFK-B, and its promoter region has multiple copies of the Sp1 and AP-2 consensus sequences. Its expression is induced by various cytokines, growth factors, ROS, lipopolysaccharides, and heavy metals. The gene codes for a homotetrameric enzyme and each monomer is Mn + 3, and it has 60% homology to the *SOD3* gene [5,9].

Some of the variants identified in this gene are associated with decreased enzymatic activity, as the rs4880 *C > T* (*p*. Ala16Val, A16V, V16A, 47*C > T*) and rs5746136 *C > T* or *G > A* (c.\*441*G > A*, −102C > T) are associated with the development of premature aging [10], cardiovascular diseases [11], diabetes mellitus type 2 (DM2) [12], and cancer [9,13], among others. The rs4880 non-synonymous variant is the product of the change of *C* for *T* in codon 16 of exon 2 of the *SOD2* gene. It is characterized by the change of alanine (*GCT*) for valine (*GTT*) aa, and has been associated with a decrease (30–40%) in the activity of the SOD2 enzyme caused by the *TT* genotype. The reported frequency of the rs4880 variant depends on the population studied, and the *C* allele showed a frequency in the controls of 30–60% among European, African, and American populations, and 10–17% among Asian and Japanese populations (https://www.ncbi.nlm.nih.gov/snp/rs4880, accessed on 16 September 2022).

The rs5746136 variant is located in the 5th intron near $3'$UTR at 65 base pair downstream of the poly-A site, and < 1 kb upstream from SP1 and the NF-kB transcription element sequences. Although the biological effect of the variant is unknown, some theories have proposed that it could participate in the modulation of gene transcription, transport mRNA to the cytoplasm, and contribute to the half-life of mRNA [14,15]. Genetic variants in N6-methyladenosine are associated with bladder cancer risk in the Chinese population [16]. Variation in the reported frequency also depends on the population analyzed, and the *C* allele showed a frequency in controls of 40–80% among European, Asian, African, and Mexican populations (https://www.ncbi.nlm.nih.gov/snp/rs5746136, accessed on 16 September 2022). The *SOD2* gene variant may determine BC susceptibility, but the association studies that examined the rs4880 and rs5746136 variants and BC risk in the Mexican population remains unknown. For this reason, we consider it important to determine the frequency of the rs4880 and rs5746136 variants, and whether there is an association between *SOD2* gene polymorphisms and Mexican women with BC.

## 2. Materials and Methods

Blood samples were collected from 1174 women participating in the study, of whom 818 were patients with clinically and histologically confirmed BC and 356 healthy controls who donated blood. The local ethics committee (N°1305) at Centro de Investigación Biomédica de Occidente (CIBO) from the Instituto Mexicano del Seguro Social (IMSS) approved the study. The study groups with residents of the metropolitan area of Guadalajara, not age-matched, and with no familial samples were included. All samples from study participants were obtained after signing the informed consent. All procedures performed were in

accordance with the Helsinki Declaration. Clinical and demographical data were obtained using written questionnaires. DNA was extracted from peripheral blood lymphocytes using standard protocols [17]. The rs4880 variant was identified by real time PCR using the following probes: VIC-5′- CTGCCTGGAGCCCAGATACCCCAAA[A/G]-3′, and FAM-5′-CCGGAGCCAGCTGCCTGCTGGTGCT-3′(C_8709053_10), as designed and validated by Applied Biosystem (Thermo Fisher Scientific, Waltham, MA, USA). The reaction included a volume of 5 L (~10 ng) genomic DNA, 6.25 µl of TaqMan universal buffer, 0.32 l of VIC- and FAM TaqMan-labeled probe, and 3.43 µL of water per sample. They were placed in 96 well plates in a light-covered system and read in the ABI 7300 Real Time PCR System (Applied Biosystem/Thermo Fisher Scientific, Waltham, MA, USA). The amplification conditions were as follows: at 95 °C for 10 min followed by 40 cycles, of 92 °C for 10 s, and 60 °C for 1 min. As an internal control, 10% of the reactions were analyzed in duplicate to observe concordance in all of the analyzed samples.

The PCR amplification of the rs5746136 variant was performed by PCR using the following primers: 5′-CTCATGAGGACCCAGGTGAT-3′ and 5′-GGTTGAGGCAGCTATG GAGA-3′, as previously described [14,18]. The annealing temperature was 60 °C. The PCR product was digested with *Taq* I restriction enzyme. Alleles were distinguished on 8% (19:1) polyacrylamide gels stained with silver [19]. The fragments of 42 bp and 52 bp were identified as the *CC* genotype; the fragments of 42bp, 52bp, and 94bp as the *CT* genotype; and the fragments of 94bp as the *TT* genotype.

The Hardy–Weinberg equilibrium was tested by a goodness-of-fit Chi-square test to compare the observed genotype frequencies with the expected frequencies among the control subjects. The allele frequencies were obtained by direct counting. The association analysis by the OR and binary logistic regression analysis between the studied groups was performed using the PASW Statistic Base 18 software (Chicago, IL, USA). The SHEsis Online Version program [20] analyzed the pair-wise linkage disequilibrium (D′) and haplotype frequency.

## 3. Results

Comparative demographic data from BC patients and control individuals showed that the mean age, age at menarche, menopause status, as well as hormonal, tobacco and alcohol consumption, were statistically different in BC patients compared with the control group ($p < 0.001$) (Table 1).

**Table 1.** Demographic data for the study groups.

| | BC Patients (n = 818) | | Controls (n = 356) | | *p*-Value |
|---|---|---|---|---|---|
| Age at diagnosis (years) | | | | | |
| Mean (SD) | 54.06 | (11.51) | 43.92 | (14.07) | 0.0001 * |
| <45 years [(*n*), %] | (188) | 23.0 | (192) | 54.0 | 0.0001 |
| ≥46 years [(*n*), %] | (630) | 77.0 | (164) | 46.0 | |
| Age at menarche | | | | | |
| 8–10 years [(*n*), %] | (57) | 7.0 | (14) | 4.0 | 0.0611 |
| 11–13 years [(*n*), %] | (532) | 65.0 | (335) | 94.0 | 0.0001 |
| 14–18 years [(*n*), %] | (229) | 28.0 | (7) | 2.0 | 0.0001 |
| Menopause status | | | | | |
| Postmenopausal [(*n*), %] | (564) | 69.0 | (132) | 37.0 | 0.0001 |
| Premenopausal [(*n*), %] | (254) | 31.0 | (224) | 63.0 | |

**Table 1.** *Cont.*

|  | BC Patients [(n = 818)] |  | Controls [(n = 356)] |  | *p*-Value |
|---|---|---|---|---|---|
| Hormonal consumption |  |  |  |  |  |
| Yes [(*n*), %] | (360) | 44.0 | (100) | 28.0 | 0.0001 |
| No [(*n*), %] | (458) | 57.0 | (256) | 72.0 |  |
| Tobacco consumption |  |  |  |  |  |
| Yes [(*n*), %] | (229) | 28.0 | (85) | 24.0 | 0.1631 |
| No [(*n*), %] | (589) | 72.0 | (271) | 76.0 |  |
| Alcohol consumption |  |  |  |  |  |
| Yes [(*n*), %] | (139) | 17.0 | (75) | 21.0 | 0.1141 |
| No [(*n*), %] | (679) | 83.0 | (281) | 79.0 |  |

SD (standard deviation); * Student's *t*-test.

The clinical characteristics of BC patients are shown in Table 2. The main characteristics of BC patients were obesity (41%), ≤4 pregnancies (72%), unilateral tumor localization (95%), ductal tumor type (91%), stage III tumor (35%), positive node status (73%), luminal A (46%), Ki-67 ≥ 20% (36%), metastasis presence (29%), non-chemotherapy response (37%), and benign breast disease-uterine fibroids presence (28%).

**Table 2.** Clinical data for the breast cancer group.

|  | | | BC Patients [(n = 818)] | | |
|---|---|---|---|---|---|
|  | (*n*) | % |  | (*n*) | % |
| Family history of breast cancer |  |  | Tumor stage |  |  |
| Yes [(*n*), %] | (139) | 17.0 | I [(*n*), %] | (55) | 7.0 |
| No [(*n*), %] | (679) | 83.0 | II [(*n*), %] | (240) | 29.0 |
| Body mass index (BMI) * |  |  | III [(n), %] | (285) | 35.0 |
| 18–24.9 (normal weight) [(*n*), %] | (196) | 24.0 | IV [(*n*), %] | (238) | 29.0 |
| 25–29.9 (overweight) [(*n*), %] | (286) | 35.0 | Node status |  |  |
| ≥30 (obesity) [(*n*), %] | (336) | 41.0 | Positive [(*n*), %] | (597) | 73.0 |
| Pregnancies status |  |  | Negative [(*n*), %] | (221) | 27.0 |
| ≤4 [(*n*), %] | (589) | 72.0 | Molecular type |  |  |
| ≥5 [(*n*), %] | (229) | 28.0 | Luminal A [(*n*), %] | (377) | 46.0 |
| Miscarriage |  |  | Luminal B [(*n*), %] | (174) | 21.0 |
| Yes [(*n*), %] | (254) | 31 | Her-2 [(*n*), %] | (115) | 14.0 |
| No [(*n*), %] | (564) | 69 | Triple negative [(*n*), %] | (152) | 19.0 |
| Breastfeeding |  |  |  |  |  |
| ≤6 month [(*n*), %] | (180) | 22.0 | Ki-67 [(*n*), ≥20 %] | (295) | 36.0 |
| >6 month [(*n*), %] | (458) | 56.0 | Ki-67 [(*n*), <20 %] | (523) | 64.0 |
| No [(*n*), %] | (179) | 22.0 | Metastatic status |  |  |

**Table 2.** *Cont.*

| | | | | | (n) | % |
|---|---|---|---|---|---|---|
| | | | | **BC Patients** (n = 818) | | |
| | | (n) | % | | (n) | % |
| | Localization | | | Yes [(n), %] | (237) | 29.0 |
| | Left [(n), %] | (360) | 44.0 | No [(n), %] | (581) | 71.0 |
| | Right [(n), %] | (417) | 51.0 | Chemotherapy status | | |
| | Bilateral [(n), %] | (41) | 5.0 | Response [(n), %] | (517) | 63.0 |
| | Histology (adenocarcinoma) | | | No response [(n), %] | (301) | 37.0 |
| | Ductal [(n), %] | (733) | 90.0 | Personal medical history | | |
| | Lobular [(n), %] | (73) | 9.0 | benign breast disease-uterine fibroids ** | (228) | 28.0 |
| | Mixed [(n), %] | (12) | 1.0 | DM2-Hypertension ** | (220) | 27.0 |

* According to OMS classifications (appropriate body mass index for Asian populations and its implications for policy and intervention strategies. (Geneva, Switzerland): World Health Organization (WHO), 2004. ** On base *n* = 818.

The rs4880 variant in the *SOD2* gene was significantly different between BC patients and controls. The genotypes *CT* [odds ratio (OR) 1.5, 95% confidence intervals (CI) 1.11–2.08, *p* = 0.009], *TT* (OR 2.0, 95% CI 1.12–3.58, *p* = 0.023), *CTTT* (dominant model; OR 1.91, 95% CI 1.41–2.59, *p* = 0.001), *CCCT* (OR 2.0, 95% CI 1.12–3.58, *p* = 0.001), and *T* allele (OR 1.7, 95% CI 1.32–2.14, *p* = 0.001) were observed as risk factors for developing BC (Table 3).

**Table 3.** Genotype and allelic distribution of the rs4880, and rs5746136 variants of *SOD2* in BC patients and controls.

| Variant | | | BC | % | Controls * | % | OR | 95%(CI) | *p*-Value |
|---|---|---|---|---|---|---|---|---|---|
| **rs4880** | | **Genotype** | (n = 818) | % | (n = 211) | % | | | |
| | | CC | (330) | 40 | (119) | 56 | 0.52 | (0.38–0.07) | 0.0001 |
| | | CT | (386) | 47 | (78) | 37 | 1.5 | (1.11–2.08) | 0.009 |
| | | TT | (102) | 13 | (14) | 7 | 2.0 | (1.12–3.58) | 0.023 |
| | Dominant | CC | (330) | 40 | (119) | 56 | | | |
| | | CT + TT | (488) | 60 | (92) | 44 | 1.91 | (1.41–2.59) | 0.0001 |
| | Recessive | TT | (102) | 13 | (14) | 7 | 2.0 | (1.12–3.58) | 0.023 |
| | | CC + CT | (716) | 87 | (197) | 93 | | | |
| | Codominant | CT | (386) | 47 | (78) | 37 | 1.5 | (1.11–2.08) | 0.009 |
| | | CC + TT | (432) | 53 | (386) | 63 | | | |
| | Allele (2n = 1636) | | | | (2n = 422) | | | | |
| | | C | (1046) | 0.639 | (316) | 0.748 | 0.6 | (0.46–0.75) | 0.0001 |
| | | T | (590) | 0.361 | (106) | 0.252 | 1.7 | (1.32–2.14) | 0.0001 |
| **rs5746136** | | **Genotype** | (n = 481) | % | (n = 356) | % | | | |
| | | CC | (248) | 52 | (223) | 63 | 0.63 | (0.48–0.83) | 0.001 |
| | | CT | (193) | 40 | (118) | 33 | 1.35 | (1.01–1.80) | 0.046 |
| | | TT | (40) | 8 | (15) | 4 | 2.0 | (1.12–3.79) | 0.025 |
| | Dominant | CC | (248) | 52 | (223) | 63 | | | |
| | | CT + TT | (233) | 48 | (133) | 37 | 1.6 | (1.25–2.20) | 0.0004 |
| | Recessive | TT | (40) | 8 | (15) | 4 | 2.0 | (1.12–3.79) | 0.025 |
| | | CC + CT | (441) | 92 | (341) | 96 | | | |
| | Codominant | CT | (193) | 40 | (118) | 33 | 1.3 | (1.01–1.80) | 0.046 |
| | | CC + TT | (288) | 60 | (238) | 67 | | | |
| | Allele (2n = 962) | | | | (2n = 712) | | | | |
| | | C | (694) | 0.717 | (564) | 0.792 | 0.66 | (0.52–0.83) | 0.0004 |
| | | T | (274) | 0.283 | (148) | 0.208 | 1.5 | (1.20–1.89) | 0.0004 |

OR (odds ratio), CI (confidence intervals, *p*-value (significant < 0.05). * Hardy-Weinberg equilibrium in controls for rs4880 (chi-square test = 0.442, *p* = 0.5058), and rs5746136 (Chi-square test = 2.1; *p* = 0.1439). From sample total of controls (*n* = 356), only the genotypes for rs4880 (*n* = 211), and rs5746136 (*n* = 356) of *SOD2* variants were obtained.

The genotype distribution of the rs5746136 variant in the *SOD2* gene also showed significant differences between the BC patients and control groups. The genotypes *CT* (OR 1.3, 95% CI 1.01–1.80, *p* = 0.046), *TT* (OR 2.0, 95% CI 1.12–3.79, *p* = 0.025), *CTTT* (OR 1.6, 95% CI 1.25–2.20, *p* = 0.0004), *CCCT* (OR 3.86, 95% CI 1.79–8.36, *p* = 0.0002), and *T* allele (OR 1.5, 95% CI 1.20–1.89, *p* = 0.0004) were observed as risk factors for developing BC (Table 4). The rs4880 and rs5746136 variants in the *SOD2* gene were in the Hardy–Weinberg equilibrium in the control group (Table 3).

**Table 4.** Association of the rs4880, and rs5746136 variants of *SOD2* with age, menarche age, and hormonal consumption in the BC patients and the controls.

| Variant | Genotype | Variable | OR | 95%(CI) | *p*-Value |
|---|---|---|---|---|---|
| rs4880 | *CT* | ≤45 years old | 1.7 | (1.05–2.74) | 0.038 |
| | *CT* | 11–13 years old menarche | 1.5 | (1.13–2.21) | 0.008 |
| | *CT* | Hormonal consumption | 1.9 | (1.06–3.5) | 0.040 |
| rs5746136 | *CT* | 11–13 years old menarche | 1.55 | (1.12–2.13) | 0.008 |

OR (odds ratio), CI (confidence intervals, *p*-value (significant < 0.05).

The comparison of the rs4880 and rs5746136 variants of the *SOD2* gene frequency in the Mexican controls with other control populations from different ethnic groups is shown in Figure 1.

Significant differences were observed in genotype *CT* when comparing the rs4880 variant stratified by age (≤45 years old), hormonal consumption and menarche age (11–13 years old). Significant differences are also shown for both rs4880 and rs5746136 variants when compared with BC and control groups (Table 4).

**(A)**

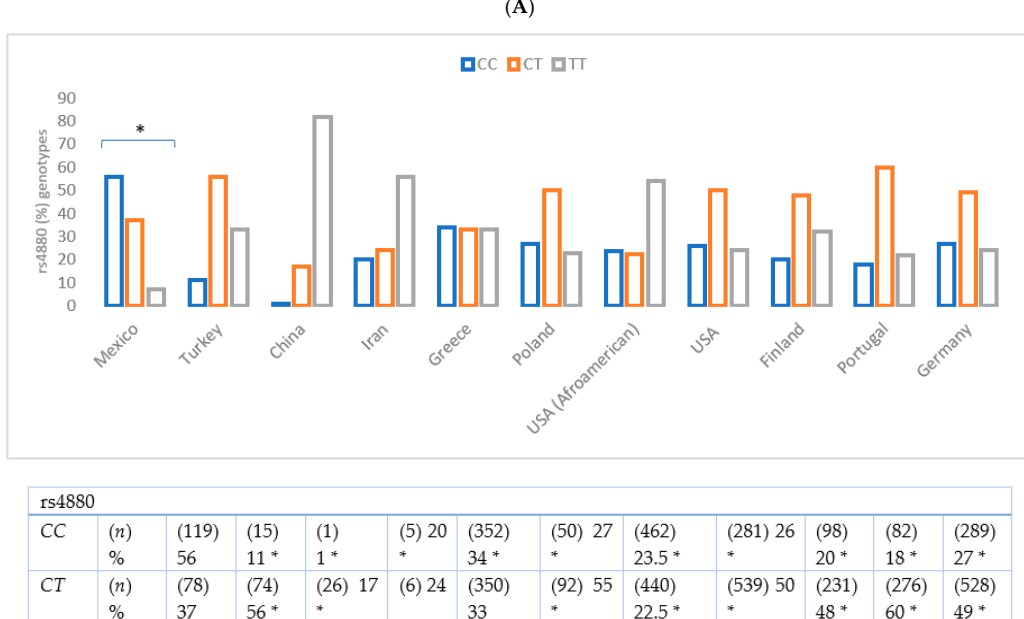

| rs4880 | | | | | | | | | | | | |
|---|---|---|---|---|---|---|---|---|---|---|---|---|
| CC | (n) | (119) | (15) | (1) | (5) 20 | (352) | (50) 27 | (462) | (281) 26 | (98) | (82) | (289) |
| | % | 56 | 11 * | 1 * | * | 34 * | * | 23.5 * | * | 20 * | 18 * | 27 * |
| CT | (n) | (78) | (74) | (26) 17 | (6) 24 | (350) | (92) 55 | (440) | (539) 50 | (231) | (276) | (528) |
| | % | 37 | 56 * | * | | 33 | * | 22.5 * | * | 48 * | 60 * | 49 * |
| TT | (n) | (14) 7 | (44) | (126) | (14) | (343) | (41) 23 | (1053) | (264) 24 | (153) | (99) | (263) |
| | % | | 33 * | 82 * | 56 * | 33 * | * | 24 * | * | 32 * | 22 * | 24 * |

**Figure 1.** *Cont.*

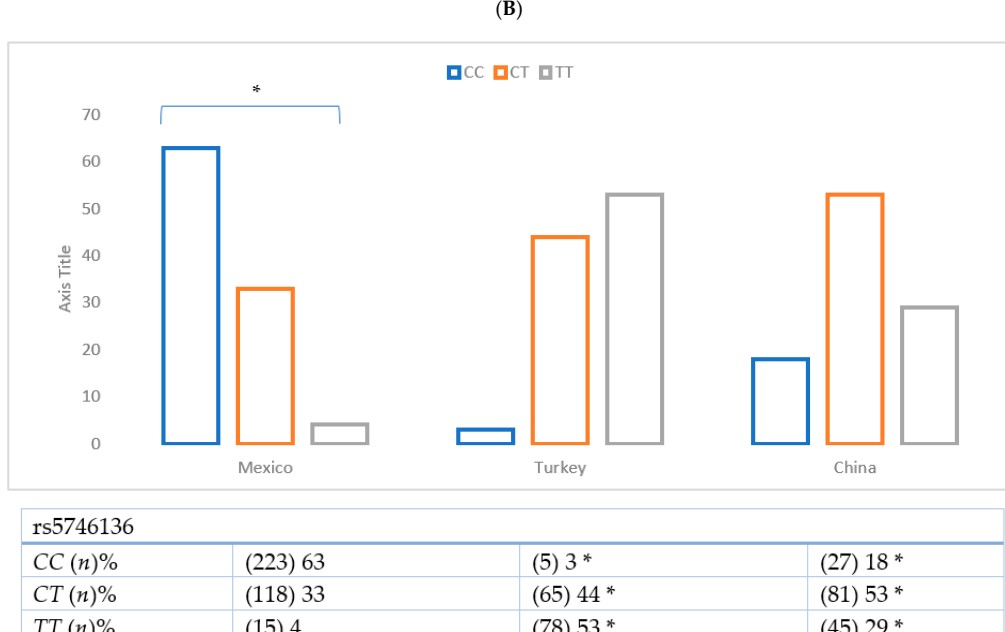

| rs5746136 | | | |
|---|---|---|---|
| CC (*n*)% | (223) 63 | (5) 3 * | (27) 18 * |
| CT (*n*)% | (118) 33 | (65) 44 * | (81) 53 * |
| TT (*n*)% | (15) 4 | (78) 53 * | (45) 29 * |

**Figure 1.** Frequency comparison of the rs4880 (**A**) and rs5746136 (**B**) variants in the *SOD2* in the Mexican women control with other populations. When comparing the frequency of the *CC*, *CT* and *TT* genotypes of the rs4880 and rs5746136 variant of the Mexican population with other populations, significant differences were observed (*p* < 0.001). * Data from present paper.

The association of clinical characteristics with the rs4880 and rs5746136 variants of *SOD2* in the BC group is shown in Table 5. The BC patients who had the dominant model *CTTT* genotypes of the rs4880 *SOD2* variant and displayed obesity (OR 3.7, 95% CI 1.07–12.9, *p* = 0.039), presence of DM2-SAH (OR 1.6, 95% CI 1.05–2.52, *p* = 0.028), and a non-chemotherapy response (OR 1.6, 95% CI 1.97–1.27, *p* = 0.002) showed a risk of developing BC. In addition, BC patients with the *TT* and *CT* genotypes of the rs5746136 variant showed BC susceptibility with Ki-67 (≥20%) (OR 2.9, 95% CI 1.16–7.36, *p* = 0.022) and luminal A type (OR 16, 95% CI 0.46–0.21, *p* = 0.041), respectively, and the dominant model *CTTT* for a chemotherapy partial response (OR 2.37.5, 95% CI 1.06–5.32, *p* = 0.035).

**Table 5.** Association of the rs4880, and rs5746136 variants of *SOD2* with clinical variables of BC patients.

| Variant | Clinical Variable | OR | 95% (CI) | *p*-Value |
|---|---|---|---|---|
| rs4880 | Obesity (BMI 30–40) | 3.7 | (1.07–12.9) | 0.039 |
| | DM-SAH * | 1.6 | (1.05–1.27) | 0.002 |
| | Non-chemotherapy response | 1.6 | (1.97–1.27) | 0.002 |
| rs5746136 | Ki-67 (≥20%) | 2.9 | (1.16–7.36) | 0.022 |
| | Luminal A | 1.6 | (0.46–0.21) | 0.041 |
| *CTTT* | Chemotherapy partial response | 2.37 | (1.06–5.32) | 0.035 |

OR (odds ratio), CI (confidence intervals, *p*-value (significant < 0.05). *: Personal medical history. Chemotherapy treatment with Anthracyclines (e.g., doxorubicin, epirubicin, liposomal doxorubicin), taxanes (docetaxel, paclitaxel), cyclophosphamide, capecitabine and trastuzumab were evaluated according to the pathological Ryan's classification, described as follows: 1. Moderate response (single cells or small groups of cancerous cells), 2. Minimum response (residual cancer surrounded by fibrosis), and 3. Poor response (minimal or no tumor destruction, extensive residual cancer).

The rs4880 and rs5746136 variants of *SOD2* showed no linkage disequilibrium (D′ = 0.265 and r′ = 0.050) in the control group.

Table 6 shows the haplotype frequency and association of the *SOD2* variant with BC. The most frequent haplotype was *C/C* (51% and 57%), followed by *T/C* (21% and 17%); however, no statistical differences were observed between the patients and the controls, respectively. Nonetheless, evident differences were observed with the haplotype *T/T* (16% and 9%; *p* < 0.01).

**Table 6.** Haplotype frequency of the rs4880, and rs5746136 variants of *SOD2* in BC patients and controls.

| *SOD2* Gene | | Patients | | Controls | | | |
|---|---|---|---|---|---|---|---|
| **rs4880** | **rs5746136** | *n* | % | *n* | % | **OR95%(CI)** | *p*-Value |
| *C* | *C* | (474) | 51 | (128) | 57 | 0.779 (0.58–1.04) | 0.095 |
| *C* | *T* | (114) | 12 | (37) | 17 | 0.693 (0.46–1.03) | 0.072 |
| *T* | *C* | (197) | 21 | (39) | 17 | 1.250 (0.85–1.82) | 0.245 |
| *T* | *T* | (149) | 16 | (19) | 9 | 1.986 (1.20–3.26) | 0.005 |

Finally, there was a genotype combination association between BC patients; the *CTCC* was associated with risk susceptibility to the presence of DM2-SAH (OR 1.9, 95% CI 1.09–3.4, *p* = 0.024), which was shown to be associated with BC risk.

## 4. Discussion

In the present study, we show a brief description of the clinical characteristics of a group of Mexican women with BC. It is especially considered a health problem in women around 50 years of age, who are economically productive and who play an important role in their families [3,5,21]. In this group of women, around 20 to 45% have BC risk factors, such as age ≥ 45 years, menarche early and late age presence, hormone and tobacco use, family history of BC, personal history of fibroadenomas and uterine myomatosis, as well as DM2 and SAH, less than four gestations, breastfeeding their children for less than 6 months, miscarriage, and obesity. In addition, in most of the highlighted patients, the presence of the tumor is unilateral, ductal, in late stages, with positive lymph nodes, luminal A or B, with Ki-67 less than 20%, without metastasis to other organs, and with a good response to chemotherapy. In this regard, the demographic clinical characteristics presented by this group of patients should be considered as being important points of etiology regarding the evolution of the tumor of patients in the Mexican population [2,13]. In recent years, important campaigns for early detection and care have been carried out; however, this effort has not yet been sufficient, since late detection and limited access to medical care continue to be a health problem in our country [2,3,5,13]. Therefore, more studies are needed in order to understand the genomics of the Mexican population, and to gain a better understanding of the biological mechanisms of diseases that are considered multifactorial, including BC [3,5,13].

In this study, we observed differences in the age of menarche, tobacco and alcohol use, and hormones consumption between BC patients and the controls. The relationship between these two factors is well established [22–24]. In this sense, we observed statistically significant differences when comparing the group of patients with BC vs the control group. These differences are due to intrinsic factors of the study groups. In addition, the comparative analysis of the *CT* genotype of the rs4880 and rs546136 variants showed an association of risk to BC; this may also be due to the factors of each study group.

Different theories have been proposed about the function of the SOD2 enzyme in the regulation of oxidative stress that ROS generates (tumor promoter) in the cell, and the development of cancer. Additionally, the *SOD2* gene has binding sites for different transcription factors that act as a ligand to activate the transcription and participate as a cell defense system against agents that induce oxidative stress [9].

Association studies in relation to the rs4848 and rs5746136 variants of *SOD2* and BC have shown different types of susceptibilities; some with risk [25–28], others with protection [29], and still others without an association [30–33]. However, little is known about the association of the rs4848 and rs5746136 variants of *SOD2* and the BC gene in Mexican BC patients. In the present study, the frequency of *CT, TT, CTTT* (dominant model) genotypes, and the T allele of both rs4880 and rs5746136 variants of *SOD2*, showed statistically significant differences between BC patients and the controls ($p < 0.05$), and were associated with the risk of developing BC.

This is the first study where the association between the rs4848 and rs5746136 variants of *SOD2* as they relate to BC in a Mexican population is analyzed. Although a previous study found that the *T* allele was associated with the development of microalbuminuria in DM2 patients in the Mexican population [12], similar results were observed with respect to the distribution of genotypes in the Mexican population. This data supports the hypothesis that the *C* allele (alanine) of the rs4880 variant confers a positive susceptibility to BC and, conversely, the *T* allele that codes for valine has the effect of susceptibility to BC risk. A plausible explanation is that the conformational configuration of valine reduces the efficiency of the activity of the protein in the mitochondria, and therefore, an accumulation of ROS is produced, causing oxidative stress and damage to the ductal or lobular cells of the breast tissue [5,9]. Thus, mutations, epigenetic modifications, promoter methylation, and epigenetic cytosine methylation in the *SOD2* gene promoter may likely have an impact on *SOD2* regulation in BC cells, resulting in the reduced expression of *SOD2* [9].

Association studies on the rs5746136 variants of *SOD2* and BC remain unknown. However, this variant has been associated with a susceptibility of increased risk in smokers with gastric cancer [34] and with polycystic ovary syndrome [35]. It is also associated with a susceptibility of bladder cancer [16].

In this case, various studies have analyzed the expression of *SOD2* in BC; however, the regulatory mechanisms in the development of BC still need to be understood. The T alleles from *SOD2* in both (rs4848 and rs5746136) variants likely have a deficient activity effect on the SOD2 enzyme. As a result, the cellular protective mechanisms and the antioxidant defense capacity are inefficient. The oxidative stress is generated; therefore, gene regulatory mechanisms can anticipate and initiate the development of BC [4].

When conducting the comparative analysis of the rs4848 and rs5746136 variants of *SOD2* in the group of Mexican women (the control group with the control groups of other populations), the distribution genotypes showed statistically significant differences with populations from Turkey, China, Iran, Greece, Poland, USA (African Americans), USA (Caucasian), Finland, Portugal, and Germany. There was an exception for the *CT* genotype of rs4880 from Iranian and Greek populations. The distribution genotypes showed statistically significant differences with the Turkish and Chinese populations with respect to the Mexican population, pointing to the genetic heterogeneity of these polymorphisms in other populations (https://www.ncbi.nlm.nih.gov/snp/rs4880, and /snp/rs5746136; access on 12 September 2022).

The data available for other population groups indicate high allele frequencies for these variants. It is clear that the *T* allele is more common in the rest of the populations compared to ours; for instance, we observe that reports from China have a frequency of 82% of the rs4880 variant, and Turkey has a frequency of 53% for the rs5746136 variant. Furthermore, the frequency of genotypes observed for both variants is similar to HapMap for the Mexican American sample from Los Angeles, California (https://www.ncbi.nlm.nih.gov/snp/rs4880 and rs5746136; access date on 12 September 2022). Data corroborated a study with DM2 in the Mexican population, in which the T allele was associated with the development of microalbuminuria in DM2 patients [12].

In our study, the *CT* heterozygous genotype of the rs4880 variant was associated with risk factors in BC stratified by age (≤45 years old), hormonal consumption and age of menarche (11–13 years old). Additionally, the presence of the *CT* (heterozygous) genotype of the rs5746136 variant and age of menarche (11–13 years old) were seen as risk factors

for BC. Moreover, the analyzed association of the rs4848 and rs5746136 variants of the SOD2 gene in Mexican BC patients in our study, demonstrated that the dominant model of the *CTTT* genotype of the rs4880 variant was a risk factor for BC susceptibility stratified by different clinical pathology parameters, such as menopause, obesity (BMI 30–40), the presence of DM2-SAH, and a non-chemotherapy response. Regarding the rs5746136 variant, it is observed that the *TT, CT*, and *CTTT* genotypes were associated with susceptibility to BC with Ki-67 ($\geq$20%), luminal A, and a partial response to chemotherapy, respectively. In this sense, different regulatory mechanisms have been proposed for the development of cancer [9]. It has been shown that, at a younger age, the concentration of total SOD in serum is higher, and individuals carrying the *TT* genotype of the rs4880 variant had a two-fold increased risk of developing obesity. In addition, it has been proposed that this polymorphism might be a hereditary factor for developing obesity, leading to increased susceptibility to oxidative damage in tissues such as $\beta$ pancreatic cells, and altering the response of these cells to the glycemic status. It is likely that the Val isoform of SOD2 may lead to a decreased resistance against ROS produced in the mitochondria, and oxidative damage to proteins caused by less efficient SOD2 transport into the mitochondria [36]. We also found that the age of menarche was associated with a risk of BC, indicating the importance of puberty. The increase in height between 8 and 14 years of age conferred a higher risk of BC. Within the past century, adult height and the prevalence of obesity have increased and the age at menarche has decreased, indicating that changes in some environmental conditions are important and likely interact with genetic factors. However, the estrogens that adipose tissue produced may promote differentiation of the breast epithelium, which produces oxidative stress. This has shown that the decreased antioxidant enzyme activity of the SOD family produces DNA damage by oxidative stress, which could cause cancer [37].

Regarding the association of Ki-67, considered as a proliferation marker of BC and other types of cancer, previous studies have revealed that antioxidant enzymes as SOD are closely linked to an increased cell proliferation in tumors. Therefore, there are many intrinsic factors in tumors which produce oxidative stress and which damage DNA that give rise to carcinogenesis [4].

Anti-neoplastic agents like anthracyclines and taxanes (including fluorouracil, cyclophosphamide, epirubicin, and capecitabine) are essential to adjuvant therapy in treating BC; however, they have been shown to induce excessive ROS production and, consequently, cell apoptosis. SOD2 is an important regulator involved as an antioxidant agent in the ROS metabolic processes, which may further interfere with the drug-resistant signaling pathways. Previous studies have reported the rs4880 variant of *SOD2* to be significantly associated with drug-induced hepatotoxicity [38]. Genotype TT has demonstrated that it increases the risk of hepatotoxicity in leukemic patients, as was observed in the findings in the adult Hispanic population [39].

In addition, the haplotype and genotype combinations' association of rs4848 and rs5746136 variants of the *SOD2* gene were determined between BC patients and the control groups. The haplotypes showed no linkage disequilibrium with each other. We observed that the *T/T* (OR 1.98; 95% CI 1.20–3.26.27, *p* = 0.005) haplotype was associated with susceptibility to BC. Moreover, we determined that the genotype combination association was a risk factor for developing BC stratified by different clinical pathology parameters, the *CT/CC* with presence of DM2-SAH, and a non-chemotherapy response by recurrence. To our knowledge, this is the first study in BC patients of the Mexican population to report this association; however, we could elucidate that the progression of cancer is associated with adverse clinical outcomes, and it may modify the expression of different molecular factors, including stress oxidative mechanisms, which could alter the regulation of cellular processes [5,8,9,39].

On the other hand, the progression of cancer is not only related to the monogenic inheritance of a protein variant, but is multifactorial, depending on the interaction of several genes involved in multiple metabolic pathways, epigenetic events, and environmental factors [3].

## 5. Conclusions

Our results showed that the rs4848 and rs5746136 variants of the *SOD2* gene were associated with BC risk when comparing controls and BC patients for the codominant, recessive, and co-dominant models and T alleles, respectively. There were evident differences observed in *CT* genotypes in patients and the controls with age $\leq 45$ years old, age (11–13 years old) of menarche, and hormonal consumption for the rs4880 variant. Additional differences were also observed in patients with the *CTTT* (the dominant model) genotype and obesity, a personal history of DM2-SAH, and a non-chemotherapy response to the rs4880 variant. For those with the rs5746136 variant, differences were observed in Ki-67 ($\geq 20\%$), luminal A type BC, and a chemotherapy partial response in BC patients with the *TT*, *TC*, and *CTTT* genotype, respectively. The haplotype *T/T* was observed as a risk factor for BC. The allele combination was associated with the *CTCC* genotype and personal history of DM2-SAH. New results confirm previous data that these factors significantly contribute to BC susceptibility in the analyzed sample from a Mexican population; however, further studies are required in order to confirm these observations.

**Author Contributions:** M.P.G.-A., R.R.-P., J.Y.S.-L., E.A.M.-F. and M.A.R.-R.; methodology, M.P.G.-A., J.Y.S.-L., B.C.G.-M., D.P.P.-V., A.F.G.-R. and J.I.D.-S.; software, M.P.G.-A., L.E.F., N.A.M.-D. and G.M.Z.-G.; validation, M.P.G.-A., L.E.F., G.M.Z.-G., A.M.P.-P. and B.C.G.-M.; formal analysis, M.P.G.-A., D.P.P.-V., A.F.G.-R. and L.E.F.; investigation, M.P.G.-A., R.R.-P. and M.L.L.-V.; resources, M.P.G.-A., A.M.P.-P., B.C.G.-M. and G.M.Z.-G.; data curation, M.P.G.-A., L.E.F., G.M.Z.-G., B.C.G.-M. and A.M.P.-P.; writing—original draft preparation, M.P.G.-A., L.E.F. and G.M.Z.-G.; writing—review and editing, M.P.G.-A., L.E.F. and G.M.Z.-G.; funding acquisition, M.P.G.-A. All authors have read and agreed to the published version of the manuscript.

**Funding:** This research was financially supported through of Proyecto 320484, Ciencia Básica y/o Frontera, Modalidad: Paradigmas y Controversias de la Ciencia 2022, CONACYT, Fundación IMSS and CIBO, IMSS, grant number 320484.

**Institutional Review Board Statement:** The study was conducted in accordance with the Declaration of Helsinki and approved by the Institutional Review Board (Ethics Committee 13058) of Centro de Investigación Biomédica de Occidente (protocol code R-2021–1305-006 and November 2021 of approval) for studies involving humans.

**Informed Consent Statement:** Informed consent was obtained from all subjects involved in the study.

**Data Availability Statement:** Data that support the findings reported in this paper are available from the corresponding author upon reasonable request.

**Acknowledgments:** This research was financially supported through of Proyecto 320484, Ciencia Básica y/o Frontera, Modalidad: Paradigmas y Controversias de la Ciencia 2022, CONACYT, Fundación IMSS and CIBO, IMSS grants.

**Conflicts of Interest:** The authors declare no conflict of interest.

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
