# Peer review of "SOD2 Gene Variants (rs4880 and rs5746136) and Their Association with Breast Cancer Risk"

_cimb, doi:10.3390/cimb44110355_

Round 1

Reviewer 1 Report (New Reviewer)

In this study, that I consider well written, only one or other error should be corrected (Ex: line 191- chemoterapy response partial to "chemoterapy partial response", there is for me however, a fundamental problem:

The characteristics of the control group that may determine some important bias in the study as.the two groups (control and Cancer ) have differences in relation to non-genetic risk factors such as the number of menopausal women or the number of women with hormone therapy. Those differences may  influence the results, and makes it unsure whether the differences found are really due to the genetic factors. I am refering  fundamentally to tables 1, 3 and 4.

I am of the opinion that the results should be analysed and discussed taking into account this factors

Author Response

Guadalajara, Jalisco. October 12, 2022

Ms. Luna Liu

Assistant Editor

E-Mail: [email protected]

We appreciate your attention to our manuscript entitled “Title SOD2 gene variants (rs4880 and rs5746136) and their association with breast cancer risk” by Martha P. Gallegos-Arreola, Ramiro Ramírez-Patiño, Josefina Y. Sánchez-López, Guillermo M. Zúñiga-González, Luis E. Figuera, Jorge I. Delgado-Saucedo, Belinda C. Gómez-Meda, Mónica A. Rosales-Reynoso, Ana M. Puebla-Pérez, María L. Lemus-Varela, Asbiel F. Garibaldi-Ríos; Nayely A. Marín-Domínguez; Diana P. Pacheco-Verduzco and Emaan A. Mohamed-Flores. We are submitting to your editorial authority for publication in

Current Issues in molecular biology.

All comments from reviewers were considered and pertinent changes were included in this version of the manuscript. The changes made to the document are detailed below.

Reviewers 1

In this study, that I consider well written, only one or other error should be corrected (Ex: line 191- chemoterapy response partial to "chemoterapy partial response", there is for me however, a fundamental problem:

“The characteristics of the control group that may determine some important bias in the study as.the two groups (control and Cancer ) have differences in relation to non-genetic risk factors such as the number of menopausal women or the number of women with hormone therapy. Those differences may  influence the results, and makes it unsure whether the differences found are really due to the genetic factors. I am refering  fundamentally to tables 1, 3 and 4.”

“I am of the opinion that the results should be analyzed and discussed taking into account this factors”

The manuscript was revised once more, the grammatical errors have been corrected and are highlighted in yellow in the manuscript. Comments regarding tables 1 and 4 were included. Table 3 only describes the genotypic and allelic frequency of each studied variable.

Reviewer 2 Report (New Reviewer)

The authors have concisely explained that 2 gene variants of SOD gene and their association with breast cancer. However, there are several sections of the introduction, materials and methods, discussion (refer file attached) are derived from previously published materials which makes me question the scientific integrity of the study.

Author Response

Reviewers 2

“The authors have concisely explained that 2 gene variants of SOD gene and their association with breast cancer. However, there are several sections of the introduction, materials and methods, discussion (refer file attached) are derived from previously published materials which makes me question the scientific integrity of the study,”

Although the reviewer's comment is accurate, we must clarify that the article he refers to is from our group, and we have worked on this topic for more than 15 years. Due to the aforementioned, it is not uncommon that there is a lot of coincidence in the general information included in the document; also the groups of patients and control groups were analyzed (totally or partially) in other studies; however, some variations are included in the wording, without altering the meaning to be expressed. What if we want to make it clear that the data/results presented related to the 2 variants of the SOD gene are totally new and original, which I think should clear the reviewer's doubts about the scientific integrity of the work. 

Reviewer 3 Report (New Reviewer)

Line 62 (and elsewhere):  H2O2  (use subscripts)

Line 92:  "...determine the frequency,...."   frequency of what?

Line 148:  If the name of the organization (WHO) is in English, the location (Ginebra, Suiza)) should also be in English (Geneva, Switzerland).

Figure 1:  Font is very small (difficult to read at 100%).

Line 217:  "families" might be better than "family".

Line 237: "promotor tumor" should probably be changed to "tumor promotor".

Lines 348 through 354:  Looks out of place and should be deleted.

Line 367: "The previous evidence confirms...".  Usually, new data confirm previous data, not the other way around.

References:  References should be presented in a consistent format.  In the current version, some titles are in lower case while others are capitalized.

Author Response

Reviewers 3

Comments and Suggestions for Authors

Line 62 (and elsewhere):  H2O2  (use subscripts)

Line 92:  "...determine the frequency,...."   frequency of what?

Line 148:  If the name of the organization (WHO) is in English, the location (Ginebra, Suiza)) should also be in English (Geneva, Switzerland).

Figure 1:  Font is very small (difficult to read at 100%).

Line 217:  "families" might be better than "family".

Line 237: "promotor tumor" should probably be changed to "tumor promotor".

Lines 348 through 354:  Looks out of place and should be deleted.

Line 367: "The previous evidence confirms...".  Usually, new data confirm previous data, not the other way around.

References:  References should be presented in a consistent format.  In the current version, some titles are in lower case while others are capitalized.

(x) English language and style are fine/minor spell check required 

Grammatical errors have been corrected and are highlighted in yellow in the manuscript. the letter of figure 1 was changed for easier reading.

Round 2

Reviewer 1 Report (New Reviewer)

Although I consider that the control group used in the study is not the ideal one and that introduces a certain bias at the results , after the safeguards and reviews made by the authors, I believe that the article should be accepted for publication

Reviewer 2 Report (New Reviewer)

The authors have addressed the issue that I raised during the revision. I agree with publication of this manuscript.

This manuscript is a resubmission of an earlier submission. The following is a list of the peer review reports and author responses from that submission.

Round 1

Reviewer 1 Report

The manuscript submitted by Gallegos-Arreola et al, ‘SOD2 gene variants (rs4880, and rs5746136) and their association with breast cancer risk’ reports the association of gene variants of SOD2 (superoxide dismutase 2) rs4880 and rs5746136 with breast cancer patients. The Manuscript is written poorly and there are many grammatical mistakes. Data presentation is poor with unnecessary long discussion. Authors must rewrite or get help to write it and submit it again. I am pointing out some errors in the manuscript here-

1.       In abstract, the authors have written ‘rs4880 variant was determinate by PCR real time’ it should be ‘rs4880 variant was determined by real-time PCR’

2.       In abstract, ‘The CT genotype of variant rs4880 29 showed significant statistical differences in patients and controls with age ≤45 years old, and hormonal consumption (P <0.05).’ It is difficult to make any sense out of this sentence.

3.       In introduction, line 62-64, simplify the sentence ‘SOD2 uses………..water’.

4.       In the introduction, line 72-64, simplify the sentence ‘Some……others.’

5.       There are several mistakes in the last paragraph of the introduction. Simplify it and focus it on the current findings of the study.

6.       In the results section, line 98, ‘age at menarched’ should be ‘age at menarche’.

7.       Rewrite the whole result section with titles for every finding and explain every data-1. Demographic and clinical features of study groups. 2. Occurrence of SOD2 gene variants rs4880 and rs5746136 in study groups. 3. Association of SOD2 gene variants rs4880 and rs5746136 with BC patients.

Reviewer 2 Report

The authors of the study "SOD2 gene variants (rs4880, and rs5746136) and their association-2 with breast cancer risk" did an excellent job compiling the SOD gene variation statistics. The authors of the study found that rs4880 and 27 rs5746136 SOD genetic variants were associated with the risk of BC susceptibility. They demonstrated that the CT genotype of variant rs4880 is significantly associated with age and hormone intake. In addition, rs4880 variant was associated with CTTT genotype and obesity, presence of DM2-SAH, and non-chemotherapy response in patients with breast cancer. However, the availability of numerous published works on the SOD gene variation in a variety of other types of cancer and its association with BC patients of different geographical locations renders the study with low-novelty and low-interest to the reader. All together, the lack of molecular mechanism associated with the these genetic variance and low interest to the readers makes the work weak, incomplete and not suitable for the this particular journal. this work will be suited elsewhere or to the most specified journals related to the study.

Reviewer 3 Report

This is a population-based study investigating a potential association between specific polymorphisms in SOD2 and breast cancer risk in the Mexican population.  The authors found that indeed certain alleles are more frequent and patients with breast cancer then in controls. Such an association has previously been reported, but this is the first study focused on the Mexican population. The experimental approach appears straightforward, and the results are credible. Enthusiasm is reduced by the following concerns:

1.    Although some alleles were more frequent in cases, other parameters known to be associated with increased cancer risk, such as early menarche and others, were likewise more frequent  in cases, complicating the interpretation of the data and raising doubts about the weight of these polymorphisms as independent indicators of breast cancer risk.

2.    Along the same line, breast carcinoma remains primarily a hormone-driven disease. The potential impact would be increased if the association between SOD2 polymorphisms and breast cancer risk was discussed and this context.

The manuscript is not well proof-read and contains numerous grammatical errors throughout. Some sentences are simply not intelligible; for example, in lines 228 to 230, the text implies that the 2 alleles are both associated with increased cancer risk, but the word "conversely" suggests that the authors mean to say that they have opposite effects. It is strongly suggested that the others have the manuscript edited by someone who is more fluent in English.